# Neurophysiological Characterization of Posteromedial Hypothalamus in Anaesthetized Patients

**DOI:** 10.3390/brainsci12010043

**Published:** 2021-12-29

**Authors:** Jesús Pastor, Lorena Vega-Zelaya, Elena Martín-Abad

**Affiliations:** Clinical Neurophysiology, Hospital Universitario de La Princesa, Fundación Investigación Biomédica La Princesa, C/Diego de León 62, 28006 Madrid, Spain; lorenacarolina.vega@salud.madrid.org (L.V.-Z.); emabad@salud.madrid.org (E.M.-A.)

**Keywords:** deep brain stimulation, intractable aggressivity extracellular action potentials, microelectrode recording, psychosurgery, Weaver syndrome

## Abstract

Deep brain stimulation (DBS) requires a precise localization, which is especially difficult at the hypothalamus, because it is usually performed in anesthetized patients. We aimed to characterize the neurophysiological properties posteromedial hypothalamus (PMH), identified by the best neurophysiological response to electrical stimulation. We obtained microelectrode recordings from four patients with intractable aggressivity operated under general anesthesia. We pooled data from 1.5 mm at PMH, 1.5 mm upper (uPMH) and 1.5 mm lower (lPMH). We analyzed 178 units, characterized by the mean action potential (mAP). Only 11% were negative. We identified the next types of units: P1N1 (30.9%), N1P1N2 (29.8%), P1P2N1 (16.3%), N1P1 and N1N2P1 (6.2%) and P1N1P2 (5.0%). Besides, atypical action potentials (amAP) were recorded in 11.8%. PMH was highly different in cell composition from uPMH and lPMH, exhibiting also a higher percentage of amAP. Different kinds of cells shared similar features for the three hypothalamic regions. Although features for discharge pattern did not show region specificity, the probability mass function of inter-spike interval were different for all the three regions. Comparison of the same kind of mAP with thalamic neurons previously published demonstrate that most of cells are different for derivatives, amplitude and/or duration of repolarization and depolarization phases and also for the first phase, demonstrating a highly specificity for both brain centers. Therefore, the different properties described for PMH can be used to positively refine targeting, even under general anesthesia. Besides, we describe by first time the presence of atypical extracellular action potentials.

## 1. Introduction

The human hypothalamus is a complex structure composed of different nuclear groups [1]. However, in general, anatomical landmarks are not available to identify a nucleus at the millimetric range. Nevertheless, the identification of hypothalamic nuclei is particularly important to obtain a good functional outcome in deep brain stimulation (DBS) to optimize battery life and decrease secondary effects, especially considering that very sensible nuclei related to arousal, circadian rhythm or hormone release are very close. Little is known about the electrophysiological properties of the hypothalamus [2,3,4], and the features of extracellular action potentials (EAPs) obtained from this structure have not been described.

Microelectrode recording (MER) is a useful tool used during DBS surgery to identify deep brain nuclei [5,6,7,8]. When surgery is performed in the thalamus, subthalamus or globus pallidus, other physiological tests in addition to MER can be performed to identify nuclei, such as cellular responses to voluntary or passive movements, tactile stimuli or paraesthesia induced by electrical stimulation [9,10,11]. However, in the hypothalamus, these stimuli are not useful because the neural systems needed are not placed near, and the only electrophysiological help is the response to electrical stimulation [3,12,13].

Nonetheless, most of the information obtained during MER in clinical practice is restricted to the mean frequency of discharge and firing pattern (e.g., tonic, phasic or more or less paused) [4,9,14,15,16], considering the EAP as a bimodal variable (present or absent) that is represented as a vertical line without an inner structure in binary plots under the assumption that the morphology of an AP does not carry any information. We have recently demonstrated that different types of EAP can be obtained from the thalamus [8], and the differential distribution could be helpful to specifically identify every structure. However, this possibility has not been demonstrated.

Intractable aggressivity describes severe cases of unprovoked aggressive behavior typically associated with some degree of mental impairment and gross brain damage [17]. The etiology can involve genetic factors, perinatal insults, brain malformation, and postencephalitic, posttraumatic or epileptic factors. Intractable aggressivity is usually accompanied by hyperkinesia, self-aggressiveness and destructiveness of objects. Patients generally need to be institutionalized and managed with major restraining measures. Surgical treatment with posteromedial hypothalamomy was initially successful [17,18]. Deep brain stimulation (DBS) at the posteromedial hypothalamus (PMH) has widely replaced ablative procedures because the clinical benefits appear to be similar, but the treatment can be titrated and is also reversible with a low risk of complications [2,3,5,6,12,13,14,15,16,17].

The aim of our study was to describe in detail the electrophysiological properties of PMH in anaesthetized human patients treated by pathological aggressiveness to characterize the properties of EAP and the pattern of discharge from PMH. Considering that no anatomical physiological clues can be used to positively identify the PMH region, we defined PMH as the region where a maximal physiological response to electrical stimulation was obtained [5,15]. These properties were compared with those obtained from thalamic nuclei from a previously published work [8].

A table with all acronyms is placed at Appendix A (Table A1).

## 2. Materials and Methods

### 2.1. Patients

We studied four patients undergoing DBS treatment at PMH for intractable aggressivity (see Appendix A, Table A2 for clinical information). In three cases a bilateral implantation was performed, but in the fourth case, only a unilateral trajectory was completed.

### 2.2. Surgical Procedures

All the patients were operated on while under general anaesthesia using propofol (5.48 ± 0.28 mg/kg/h, (4.5, 6.2)) and remifentanil (0.12 ± 0.02 µg/kg/min, (0.1, 0.2)), maintaining a bi-spectral index between 40 and 45. Neuromuscular blockade during intubation was accomplished with cis-atracurium (0.5 mg/kg).

The hypothalamus was identified using a 1.5 T magnetic resonance imaging (MRI, General Electric^®^, Fairfield, CT, USA), and the coordinates were located stereotactically with a neuronavigator (BrainLab^®^, Feldkirchen, Germany). The coordinates were calculated by fusing the MRI image and computed tomography scan according to the Schaltenbrand-Wahren map (SW [19]). For hypothalamic DBS electrode placement, a tentative initial target was selected in the posterolateral hypothalamus according to the Sano’s triangle (*x* = 2, *y* = 0, *z* = −2). All the coordinates (in mm) refer to the mid-intercommissural AC-PC line (anterior commissure–posterior commissure). Neuronal recordings (Leadpoint^®^, Minneapolis, MN, USA) were obtained beginning 5 mm above the target and progressing in steps of 0.5 mm. MERs (FHC^®^, Cumberland, ME, USA) were obtained until 2–3 mm below the theoretical target. However, the analysis for this work were restricted only to the regions near the place of maximum physiological response to electrical stimulation (see below). Impedance was always above 900 kΩ (1696 ± 80 kΩ, (900, 2900)).

MERs were obtained through 3–4 microelectrodes separated by 2 mm. A microdrive was fixed to a stereotactic Leksell Coordinate Frame (Elekta^®^, Stockholm, Sweden). The bandwidth for spontaneous activity was 200 Hz–5 kHz, with a sample rate of 24 kHz. The notch filter was off. PMH region was identified by MER and response to electrical stimulation [3,13]. After the PMH was identified, a quadripolar DBS electrode was implanted, with a programmable stimulator placed in a pectoral or abdominal location.

We have divided the trajectory of the electrode where optimal response to electrical stimulation was obtained in three regions: (i) PMH, defined by the presence of maximum neurophysiological response to electrical stimulation (i.e., delta waves in EEG, increase in heart frequency, augmentation of ipsilateral muscle cervical tone and pupil diameter change and down ipsilateral eye deviation), also including recording 0.5 mm above and below, (ii) upper PHM (uPHM), 1.5 mm above the superior point included at PMH (therefore 3 points of recording track) and (iii) lower PMH (lPHM), 1.5 mm below the lowermost point recorded of PMH (also three points of recording track). EAP and raw traces from every region were pooled and analyzed together Although several adjacent electrodes were used to identify PMH, only the electrodes where the best response was obtained have been analyzed.

Reconstruction of the trajectory was described in detail elsewhere [20]. Briefly, to do that, we considered the real $x_{r}$ and $y_{r}$ coordinates measured by MRI performed one month after surgery and represented (scaled according to AC-PC distance) them on the last axial plane of the SW map, including the floor of the thalamus (z = 0). We defined this point as the real end-point ${\vec{r}}_{\mathrm{ep}}$. The angles used during MER trajectory (ring and arch) were the same as those measured from the final DBS electrode during the post-op MRI. Starting from ${\vec{r}}_{\mathrm{ep}}$ and considering the angles, we were able to reconstruct the real trajectory of the electrode in a three-dimensional space in 1 mm intervals. To select the nearest SW plane, we used only the trajectory corresponding to the centre electrode, which was the same electrode considered during the planning process. However, we reconstructed trajectories for all the electrodes. Frontal (F), sagittal (S) and axial (A) planes of SW atlas were digitized in individual jpg files and every file was identified by the distance to intercommissural mid-point. For every dimension, SW is a set of irregular and discrete series of planes (20 for the frontal, from Fa165 to Fp155, 20 for the axial, from Av085 to Ad160 and 17 for the sagittal, from S015 to S275). For every distance to ${\vec{r}}_{\mathrm{ep}}$, it’s easy to identify the coordinates where the central electrode is placed ($x_{\mathrm{ce}},{\;y}_{\mathrm{ce}},{\;z}_{\mathrm{ce}}$). Then, we obtain the orthogonal planes including this point and, for every plane, we select from the series of jpg files, the closest one. This algorithm was implemented in a MATLAB^®^ script. For every plane, we plotted the projection of all the microelectrodes.

### 2.3. Analysis of Extracellular Action Potentials

Data were exported as ASCII files, and analyses were performed off-line. The recordings spanned 30–90 s (72 − 216 × 10^4^ points). Raw recordings were digitally filtered at 500 Hz–5 kHz using a 6th-order Butterworth. We used zero-phase forward and reverse digital IIR filtering [8,20].

The polarity of the potentials was defined as positive (P) upward and negative (N) downward and was identified by order of appearance.

Some minor modifications have been performed regarding the previously described algorithm for analysis [8,9], although basically the structure remains similar.

Non-overlapping 0.2 s windowing was performed and power spectra (PS) were computed by means of fast Fourier transform (FFT) for every window. Mean ($\bar{\mathrm{PS}}$) and SD ($\sigma_{\mathrm{PS}}$) were calculated and high frequency artifact was identified by deviation exceeding > $\bar{\mathrm{PS}}\;\pm \;7.0\sigma_{\mathrm{PS}}$. These periods were subsequently removed.Identification of EAPs. For every trace (Figure 1A), we computed a maximum (V+) and minimum (V−) voltage threshold (in µV), defined as $V_{\;\pm \;}=\bar{V}\pm \;1.5\sigma_{V}$, where $\bar{V}$ is the mean and $\sigma_{V}$ is the standard deviation. EAPs must have two phases (depolarization and repolarization); therefore, we identified a tentative EAP when a positive/negative (P/N) phase was followed by a negative/positive (N/P) phase in a period of 0.20–0.65 ms. EAPs were defined as positive $\left(\frac{P}{\left|N\right|}>1\right)$ or negative $\left(\frac{P}{\left|N\right|}<1\right)$ according to the highest component identified. Modifications of spike definition and thresholds at this step were the most relevant.

**Figure 1 brainsci-12-00043-f001:**
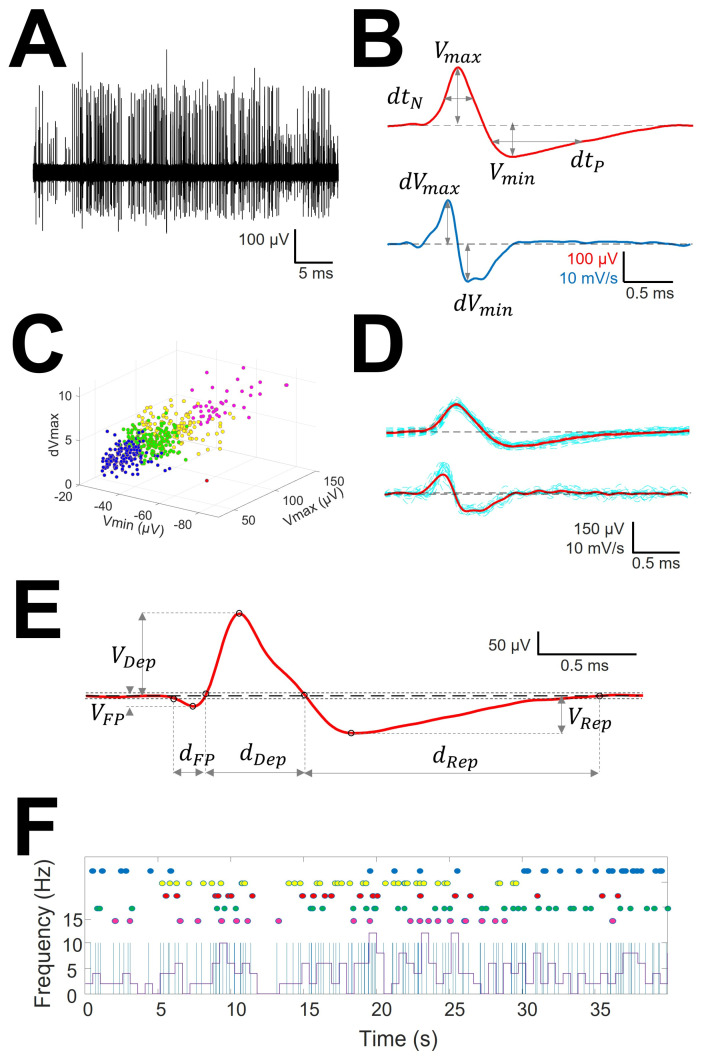
Sorting and analysis of AP and pattern discharge. (**A**) Raw recording showing EAP as vertical lines of different amplitude; (**B**) EAP (upper row) and first derivative (lower row) showing the definitions of variables computed for sorting; (**C**) clustering in different units (identified by different colors) of the raw trace. Only three dimensions are shown, although clusterization used up to 6. (**D**) mAP (upper row) and mean derivative of EAP (lower row) in red obtained from EAP and derivatives of EAP of similar morphology (blue lines); (**E**) Example of mAP showing the definition of variables analyzed. Tick dashed line represents basal voltage, and thin dotted lines represent upper and lower thresholds used to characterize the structure. Dots are the fiducial points used to define amplitudes and phases. (**F**) Analysis of raw trace. Upper dots in colors represent EAP grouped at the same mAP and therefore ascribed to the same cell. Superimposed are showed a binary plot (blue lines) where each line represents an EAP and a chart diagram showing the frequency measured at 1 s-bin.

3.Clustering was performed by an agglomerative hierarchical method, with distance between groups computed by farthest procedure. EAPs sharing similar morphologies were ascribed to the same neuron. For every EAP, we measured the maximum (V_max_) and minimum voltages (V_min_, in µV), durations of negative (dt_N_) and positive phases at half-amplitude (dt_P_ in ms), and maximum (dV_max_) and minimum values of the first derivative (dV_min_, in mV/s). These measures can be considered as a six-dimension vector for every k-EAP, ${\mathrm{EAP}}_{k}=\left\{V_{\max}^{k},V_{\min}^{k},{\mathrm{dt}}_{N}^{k},{\mathrm{dt}}_{P}^{k},{\mathrm{dV}}_{\max}^{k},{\mathrm{dV}}_{\min}^{k}\right\}$
(Figure 1B).4.Construction of the mean action potential (mAP). All the EAPs from the same cluster (Figure 1C) were averaged to obtain a canonical waveform (Figure 1D, upper row), as were the derivatives to obtain the mean derivative (1D, lower row). A minimum of 10 EAPs was averaged. The first 300 µs (72 points) of baseline were used to compute the maximum (V_AP+_) and minimum (V_AP−_) voltage thresholds (in µV), defined as $V_{\mathrm{AP}\;\pm \;}={\bar{V}}_{\mathrm{AP}}\;\pm \;2.5\sigma_{\mathrm{AP}}$, where ${\bar{V}}_{\mathrm{AP}}\;$ is the mean and $\sigma_{\mathrm{AP}}\;$ the standard deviation. We used these thresholds to identify hallmark points in mAPs (Figure 1E). Every phase can be characterized by its polarity (P/N), duration (dt_i_) and amplitude (V_i_, i = 1, 2, 3). Automatic identification of polarity of phases was done at this point (see Figure A1, (see Appendix A).

To compare the structure of different kind of mAP we have taken into account features of the first phase, i.e., duration (d_FP_) and absolute value of amplitude (|V_FP_|), the depolarization phase, i.e., duration (d_Dep_) and amplitude (V_Dep_), the repolarization phase, i.e., duration (d_Rep_) and amplitude (V_Rep_), maximum and minimum values of the first derivative (dV_max_ and dV_min_, respectively) and total duration (d_mAP_) and amplitude (V_mAP_) of mAP.

All analyses were performed in homemade MATLAB^®^R2018 scripts.

### 2.4. Analysis of the Discharge Pattern

Taking into consideration that properties of the raw recording are the most used in clinical practice, we have analyzed the following features [9,21]:
Mean frequency of the raw (${\mathrm{Freq}}_{\mathrm{raw}}$, in Hz) trace and for every neuron (${\mathrm{Freq}}_{\mathrm{cell}}$, in Hz). Both values were obtained from the inverse of the instant period.Burst Index (BI). Defined as the total number of interspikes intervals (ISI) < 20 ms, divided by the number of ISI > 20 ms. It informs about the number of bursts respecting individual discharges.$$\mathrm{BI}=\frac{{\mathrm{ISI}}_{<20\;\mathrm{ms}}}{{\mathrm{ISI}}_{>20\;\mathrm{ms}}}$$Pause Index (PI). Defined as the total number of interspikes intervals (ISI) > 100 ms, divided by the number of ISI < 100 ms. In some sense, is the opposite to BI.
$$\mathrm{PI}=\frac{{\mathrm{ISI}}_{>100\;\mathrm{ms}}}{{\mathrm{ISI}}_{<100\;\mathrm{ms}}}$$Pause Ratio (PR). Defined as the total duration of pauses (ISI > 100 ms), divided by the duration of non-pauses (ISI < 100 ms). The information given by this index is completely different from PI, because it represents the total amount of time the cell remains quiescent, compared with the duration of the burst. This ratio will be high when the refractory period of the cell after the burst would be long.
$$\mathrm{PR}=\frac{\sum_{i=1}^{N}{\mathrm{ISI}}_{<100\;\mathrm{ms}}^{i}}{\sum_{j=1}^{M}{\mathrm{ISI}}_{>100\;\mathrm{ms}}^{j}}$$Probability mass functions (pmf) of ISI for every trace. Relative frequency was computed for 10 ms bins, and the probability/bin (pi) was calculated with the following expression.$$p_{i}=\frac{f_{i}}{\sum_{i=1}^{N}f_{i}}$$where f_i_ is the frequency for the i-bin and N is the total number of bins.

These properties can be computed from the binary plot (Figure 1F).

To study the structure of these inter-spike interval histograms, we fitted the data to a double exponential decay function, using the following expression:$$y\left(x\right)=\alpha +{\beta e}^{-\gamma x}+{\delta e}^{-\varepsilon x}$$
where α, β, γ, δ and ε are the constants to be fitted. The least square sum method was used to fit the empirical data to the defined functions.

### 2.5. Statistics

Kurtosis (K) was computed for every group, and only values between 2 and 8 were acceptable for the homogeneous group [22]. Extreme outliers were removed. Statistical analysis was applied only to these groups. 

Statistical comparisons between groups were performed using the Mann–Whitney U test or one-way ANOVA on ranks if normality failed. In the last case, Dunn’s method was used for all pairwise post hoc comparisons. Normality was evaluated using the Kolmogorov–Smirnov test. Chi-square test ($\chi^{2}$) was used to assess the difference between groups and the Yate’s correction was also computed. The independence of variables (e.g., peak-to-peak action potential amplitude and amplitudes of depolarizing and repolarizing phases) was assessed by computing the rank (rnk) for the matrix containing observations (n) in rows and variables (p) in columns. Therefore, if rnk < p (considering that n > p always), then there would be some dependent variable that could be removed. However, when rnk = p, all the variables are independent and must be included in the analysis. 

SigmaStat^®^ 3.5 software (Point Richmond, CA, USA) and MATLAB were used for statistical analyses.

The significance level was set at *p* = 0.05, and the results are shown as the mean ± standard error of mean (SEM).

### 2.6. Ethics Statement

The experimental procedure was approved by the medical ethics review board of the Hospital Universitario de La Princesa and was deemed “care as usual”. Under these circumstances, written informed consent was not required because all procedures were done by clinical necessity and analysis was off-line.

## 3. Results

We analyzed traces from nine consecutive points spanning 4.0 mm from seven trajectories, including the best neurophysiological response to electrical stimulation. The presence of a raw MER between 30 and 90 s at each point ensured that we had a significant number of extracellular EAPs. Overall, we analyzed 178 cells and 53 raw traces. Considering that every mAP is composed of 34.8 ± 10.6 AP, the total number of unitary APs analyzed exceeded 6100.

### 3.1. Types of mAP According to Structure

We analyzed the structure of mAP, which is defined as the arrangement of the parts composing all the waveforms. Most of the mAPs were positive (158/178 (89%); Figure 2A–C), and few mAPs were negative (20/178 (11%); Figure 2F). Cells with atypical mAP (amAP) were recorded in 21/178 cases (12%). All these atypical cells were positive and presented structures clearly different from the other conventional types (Figure 2D,E). Negative mAPs were recorded simultaneously with positive mAPs (Figure 2G). In general, mAP had more than two phases. Only 66/178 (37%) mAP showed two phases. On the other hand, either a small positive or negative deflection was often observed before the main component (Figure 2A,B), yielding a three-phase structure. A P1N1 structure was observed in the most cases (55/178 (30.9%)) followed by a N1P1N2 structure in 53/178 (29.8%), P1P2N1 in 29/178 (16.3%), amAP in 21/178 (11.8%), N1P1 and N1N2P1 in 11/178 (6.2%) and P1N1P2 in 9/178 (5.0%).

### 3.2. Cellullar Structure of the Different Regions

To assess whether PMH regions have the same type of mAP as surrounding regions, we computed the different types of cells in the three regions analysed, i.e., uPMH, PMH and lPMH (Table 1).

A slight difference in the proportion of types of cells was noted between uPMH and lPMH ($\chi^{2}$ = 12.78, g.l = 5, *p* < 0.05), but a higher significant difference was obtained between PMH and regions above ($\chi^{2}$ = 34.89; *p* < 0.0001, g.l = 5) and below ($\chi^{2}$ = 32.46; *p* < 0.0001, g.l = 5). No differences were obtained regarding the Yate’s correction. Therefore, there is a specific proportion of cells at the PMH compared with surrounding regions. As shown in this table, there was a lower percentage of P1N1 and N1P1N2 cells and a higher percentage of P1P2N1 and amAP cells in the PMH region.

Then, we addressed whether cells of the same type (e.g., P1N1) share the same properties among the three regions. We observed limited modifications (8 modifications out of 94 possibilities −9%) across regions for the same type of cells, and all of the differences were noted in the duration. From Table 2, we can observe that for P1N1, d_Dep_ was shorter at uPMH and d_Rep_ at lPMH. P1P2N1 cells exhibited shorter d_Rep_ and d_mAP_ at uPMH. Finally, for N1P1N2 cells, d_mAP_ was longer for PMH. Therefore, the structure of mAP is conserved across the three regions for the same type of cell.

### 3.3. Tonic Properties

These features are commonly reported in the literature and are essentially the only features used in the clinical assessment during DBS surgery, even though other mathematical tools have been used in previous reports [21,23,24].

Neither of the features of the tonic discharge pattern were dependent on the region (Appendix A, Table A3). All these properties are obtained from the raw pattern discharge and are usually described by a binary plot and interspike interval histograms. Periods between consecutive EAPs can be easily computed from binary plots, where each EAP is represented by a vertical line.

The pattern does not show burst discharge; in contrast, the relatively high values of PI in the three regions denote a tonic and slightly irregular pattern of discharge. High PR values indicate an elevated refractory period. The Freq_raw_ and Freq_cell_ were low and similar to previously published data [4].

The pmf values for interspike intervals for the different regions were pooled and are shown in Figure 3. We have also computed for every pmf the mode and median. Although the mode was quite similar in the three regions, the median progressively increased from the upper to lower regions. In fact, pmf for PMH showed a small increase in probability at 90 ms (Figure 3B) and a higher increase between 90 and 130 ms for lPMH (Figure 3C). These increases were not observed for uPMH. We also fitted the descending parts of pmf to double exponential functions. The fitted probability functions for the three regions were $p_{\mathrm{uPMH}}\left(t\right)=0.017+0.091e^{-25.79t}+0.018e^{-6.12t},\;r=0.9849$, $p_{\mathrm{PMH}}\left(t\right)=0.005+0.079e^{-31.41t}+0.031e^{-5.64t},\;r=0.9944$ and $p_{\mathrm{lPMH}}\left(t\right)=0.014+0.033e^{-47.79t}+0.052e^{-7.65t},\;r=0.9824$. It is important to observe that all the correlation coefficients were greater that 0.985, and exponential coefficients were different for all three functions, implying that pmf behaves differently in the regions analyzed.

### 3.4. Atypical mAP

We recorded up to 11.8% mAP with atypical structures. These cells had amplitudes, durations or numbers of phases that clearly differed from the remainder of the mAP. We did not address a detailed characterization of this type of cell. However, obviously, we should ensure that amAP was not an artefact obtained from the superposition of phases from typical EAP. In Figure 4, we show two examples of amAP obtained from different patients demonstrating that these types of atypical EAPs are truly obtained from raw records and are not a mixture of typical action potentials. Moreover, these types of atypical EAPs have been demonstrated to be extremely relevant because they are highly concentrated in the PMH region.

### 3.5. Comparison with Thalamic Cells

An obvious question was whether mAP from the hypothalamus shares the same structural features as those recorded in thalamic nuclei. Therefore, we compared the properties of mAP with the same structure (e.g., P1N1) obtained in this work with those published by our group in a previous work [8]. These results are presented in Table 3.

For P1N1 cells, the maximum values of the first derivative were similar, and V_Dep_ and V_mAP_ were also similar. However, in hypothalamic cells, the V_Rep_ was higher, d_Dep_ was longer and d_Rep_ and d_mAP_ was shorter than that noted in thalamic cells. Additionally, the unitary frequency was higher in the hypothalamus. Regarding P1P2N1 and N1P1N2 cells, dV_max_, |dV_min_|, V_Dep_ and d_Rep_ were higher for thalamic neurons compared with hypothalamic neurons, and |V_FP_| and Freq_cell_ were also higher. However, d_FP_ was shorter for thalamic cells compared with hypothalamic cells. Finally, in the case of P1N1P2 cells, obvious polarities for V_Dep_ and V_Rep_ were opposite to the other values. In these cells, dV_max_, |dV_min_|, V_Dep_, V_Rep_, d_Rep_ and Freq_cell_ were also greater for thalamic cells compared with hypothalamic cells, but d_FP_ and d_Dep_ were longer for hypothalamic cells.

These findings indicate that mAPs of the same structure differ between both brain centers with different amplitudes and durations for depolarization and repolarization phases. In the case of P1N1P2, it is quite interesting to observe that the durations of the depolarization and repolarization phases are opposite for thalamic and hypothalamic cells, whereas d_mAP_ is similar for both regions due to compensation. Notably, the d_FP_ was always shorter for thalamic cells compared with hypothalamic cells.

Therefore, mAP is highly specific to the thalamus and hypothalamus, and this property has not been previously described in humans.

## 4. Discussion

This work demonstrates for the first time that the PMH, the region functionally defined by the maximum physiological response to electrical stimulation, has specific neurophysiological properties different from those of regions located above and below. We also noted a significant percentage of EAP located in this region with atypical structures. Notably, these EAPs with atypical morphology have not been previously reported to the best of our knowledge.

The types of mAP in the hypothalamus are similar to those described in the thalamus [8] with the majority of cells showing three phases. However, the percentage of different types of cells was different from that of thalamic nuclei, where most cells were P1P2N1 (72.7%) and less than 6.3% were P1N1. In contrast, in the hypothalamus, the most frequent structure was the P1N1 type (30.9%), whereas P1P2N1 accounted for only 16.3%. However, not only were the types of mAP dissimilar but also the properties of the same types of cells were highly different with decidedly different amplitudes and/or durations of the characteristic depolarizing and repolarizing phases. EAP can inform about several properties of the action potential intracellularly recorded [25,26]; therefore, the MER goal should not exclusively consider EAP as binary events (present/absent). To date, only EAP width has been reported for the pedunculopontine nucleus in humans, and a bimodal distribution has been observed with a longer AP attributed to cholinergic neurons and a shorter AP attributed to glutamatergic transmission [27,28]. However, no other properties have been analyzed (number of phases, features of phases, derivatives, etc.) until our previous study in thalamic nuclei [8]. Another important difference between the two brain centers is the presence of amAP in the hypothalamus. To the best of our knowledge, this is the first description of EAP that significantly differs from that conventionally described in animals and humans [4,8,26,27,28,29,30] or predicted numerically [25]. The detailed description of these cells is beyond the scope of this work, but their presence was revealed to be very important because they were concentrated at PMH. This finding can help to better identify the target. Nevertheless, it is important to keep in mind that despite their anomalous structure, they are real action potentials. However, the potential sources of origin remain unclear. It could be hypothesized that mixing different EAPs grouped together in an erroneous manner could originate these atypical structures. However, we have shown that these EAPs were truly picked up from raw records and therefore not artefacts from an erroneous sorting procedure.

Together, considering the high presence of amAP, the different combinations of types of mAP and the pattern of discharge can be used to differentiate between this region and upper and lower areas. Although the measures used to characterize the pattern of discharge (i.e., frequency, BI, PI and PR) are not region specific, the ISI pdf is. These properties can help to refine the targeting of PMH during MER in anaesthetized conditions.

From a methodological viewpoint, we must consider that all patients were under general anesthesia, which can modify neuronal discharge and conceivably synaptic function [31]. However, the exact degree of the effect remains unclear. Nevertheless, the mean frequency and other electrophysiological properties, such as local field potentials [32,33], are modified by anesthesia. Therefore, it is reasonable to identify lower values of frequency discharge in anaesthetized versus awake patients. However, neither differences in frequency nor electrical sources for local field potentials can be applied to the dissimilar values obtained regarding thalamic cells because those patients were operated on under similar conditions of general anesthesia.

The amplitude of EAP decreases monotonically with the distance from the soma [25,26,34]. This fact can explain the variation in amplitude because we can record neurons from a spherical volume of approximately 5.23 × 10^5^ µm^3^. However, the distance to the microelectrode and the net local field potential can affect the amplitude [26,30]. In fact, the total extracellular current injected into the recorded volume can affect the amount of extracellular voltage generated by an extracellular current.

As we reported for the thalamus, we observed the presence of a phase before depolarization in the majority of cells. Additionally, at the hypothalamus, we observed both opposite and similar polarities. Given that numerical and experimental data have shown the presence of capacitive current prior to the large depolarizing phase [25,30,35,36], this origin could be hypothesized for the first phase described here. However, the capacitive current is elicited by current spreading to dendrites from the soma and is always of the opposite polarity as depolarization; therefore, we must only obtain N1P1N2 and P1N1P2. Thus, capacitive current cannot explain cells such as P1P2N1 (16%) and N1N2P1 (2%). Surprisingly, the duration of this phase was consistently shorter for thalamic cells than for hypothalamic cells. This finding could be related to differences in dendrite arborization. Another possibility could be that the first phase could be related to postsynaptic potentials, which can explain the difference in polarity [8,25].

Another result difficult to explain and described in thalamic cells was the simultaneous presence of positive and negative cells. Numerous lines of experimental evidence show that depolarization is caused by the inward transmembrane current, whereas the outward current is responsible for repolarization (see [37] for a review). Mathematical simulations have shown that the shape of the EAP is proportional to the total transmembrane current of perisomatic compartments [25,30,35]. Therefore, irrespective of changes in the dendritic morphology, depolarization must always be driven by the inward Na^+^ current; consequently, they must be positive for our arrangement. However, we recorded mAPs with opposite polarities. In animal cortical recordings, a high amplitude positive action potential different from conventional negative spikes has been described, and the features of both types of spikes, such as amplitude and duration, were clearly different [38]. However, these findings were not observed in our case, where properties of positive and negative mAP (absolute magnitudes for amplitudes and phase durations) were on the same order of magnitude. As stated for amAP, we have not explained the bioelectrical sources originating from these types of cells.

Nevertheless, it is important to understand that our data were obtained from a small number of patients. Although these data seem to be robust, a larger cohort study is needed to corroborate these results.

## 5. Conclusions

In this work, we demonstrated that the neurophysiological properties of the hypothalamus are specific, and PMH can be identified during MER by the type of cells, the presence of atypical EAP and the pattern of discharge, which are different from those noted in the above and below regions. These properties can help to improve the precision of targeting during DBS surgery [39]. In addition, we described for the first time the presence of atypical EAP that does not fit either the classical depolarization-repolarization description or the three phases observed in the thalamus [8]. We demonstrated that EAP is highly specific for the recorded brain region, opening a window not only to the study of the electrophysiological properties of different cells in the processing of brain information but also to refine the identification of structures during DBS.

## Figures and Tables

**Figure 2 brainsci-12-00043-f002:**
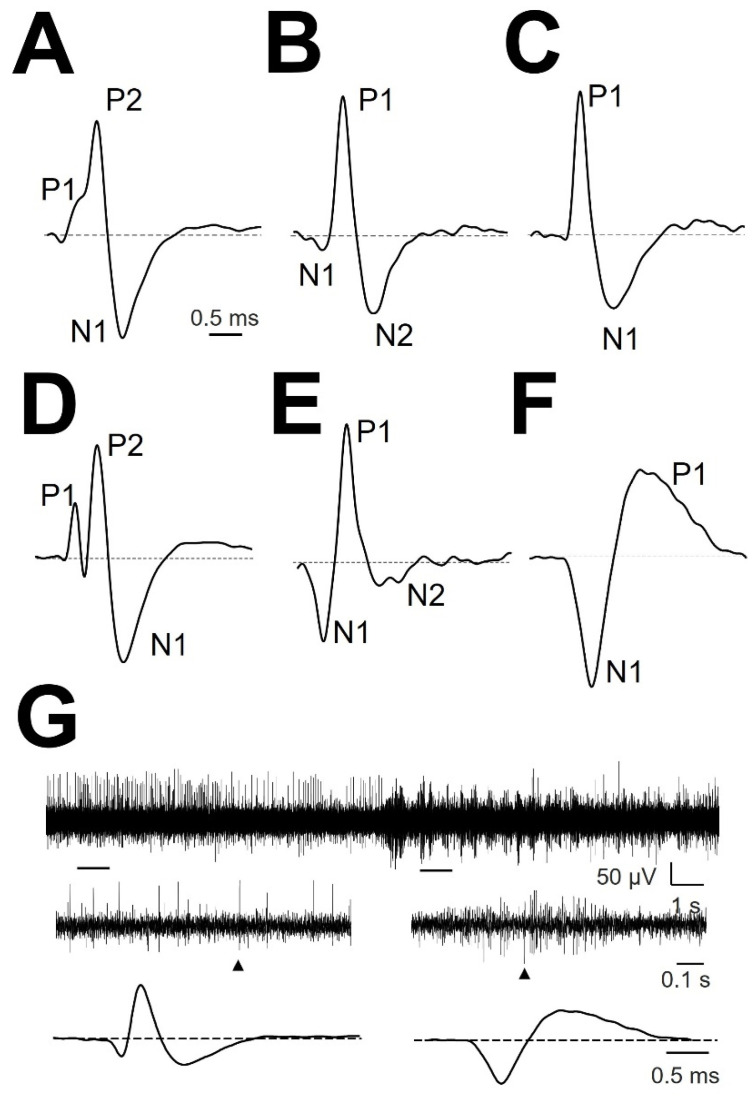
Types of normalized mAP. (**A**) P1P2N1, (**B**) N1P1N2, (**C**) P1N1, (**D**) amAP P1P2N1, (**E**) amAP N1P1N2, and (**F**) N1P1. The different phases are indicated. Dashed horizontal black line show zero voltage. (**G**) Raw trace 20 s duration from patient #3 lPMH (upper row). Two 1 s-spanned sections are enlarged (middle row), and APs obtained from everyone region are shown (lower row). Arrowheads denote the extended EAP.

**Figure 3 brainsci-12-00043-f003:**
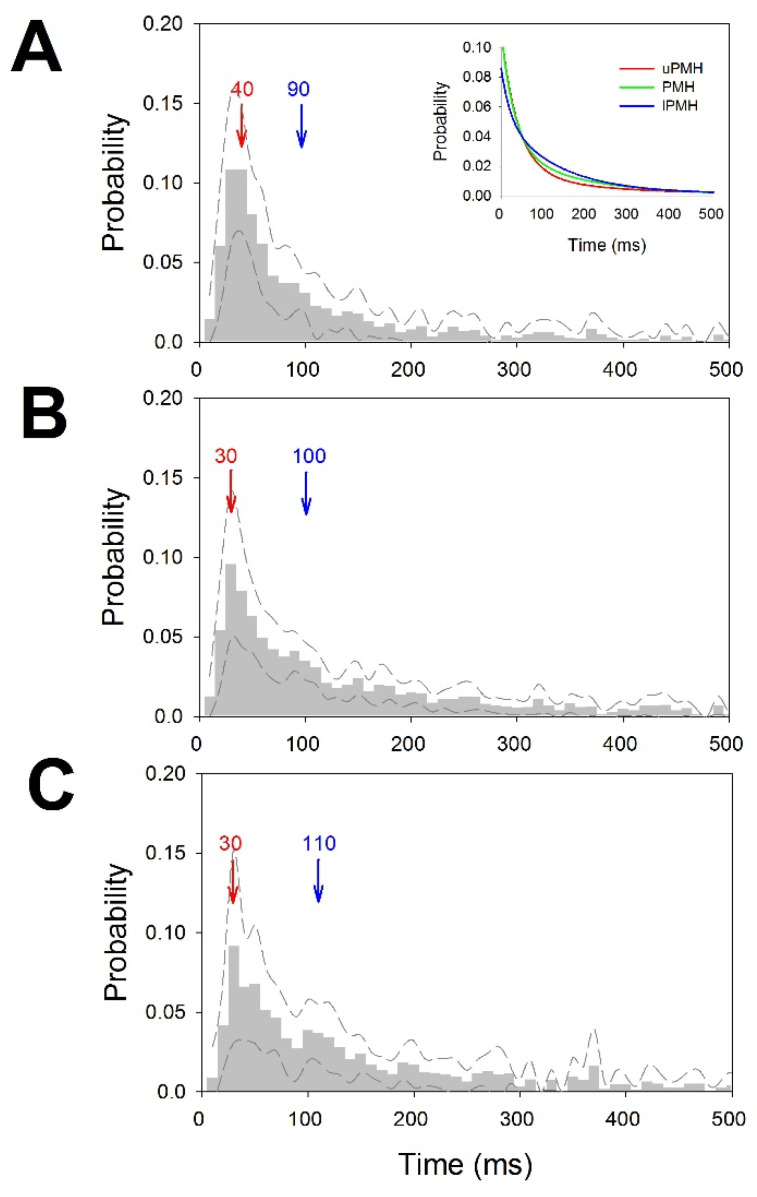
Probability mass functions for ISI distributions. (**A**) uPMH, (**B**) PMH and (**C**) lPMH. Dashed lines represent upper and lower ±2.5 × SEM. Bin = 10 ms. Mode in red and median in blue. The inset in (**A**) represents the double-exponential fitted functions for uPMH (red), PMH (green) and lPMH (blue).

**Figure 4 brainsci-12-00043-f004:**
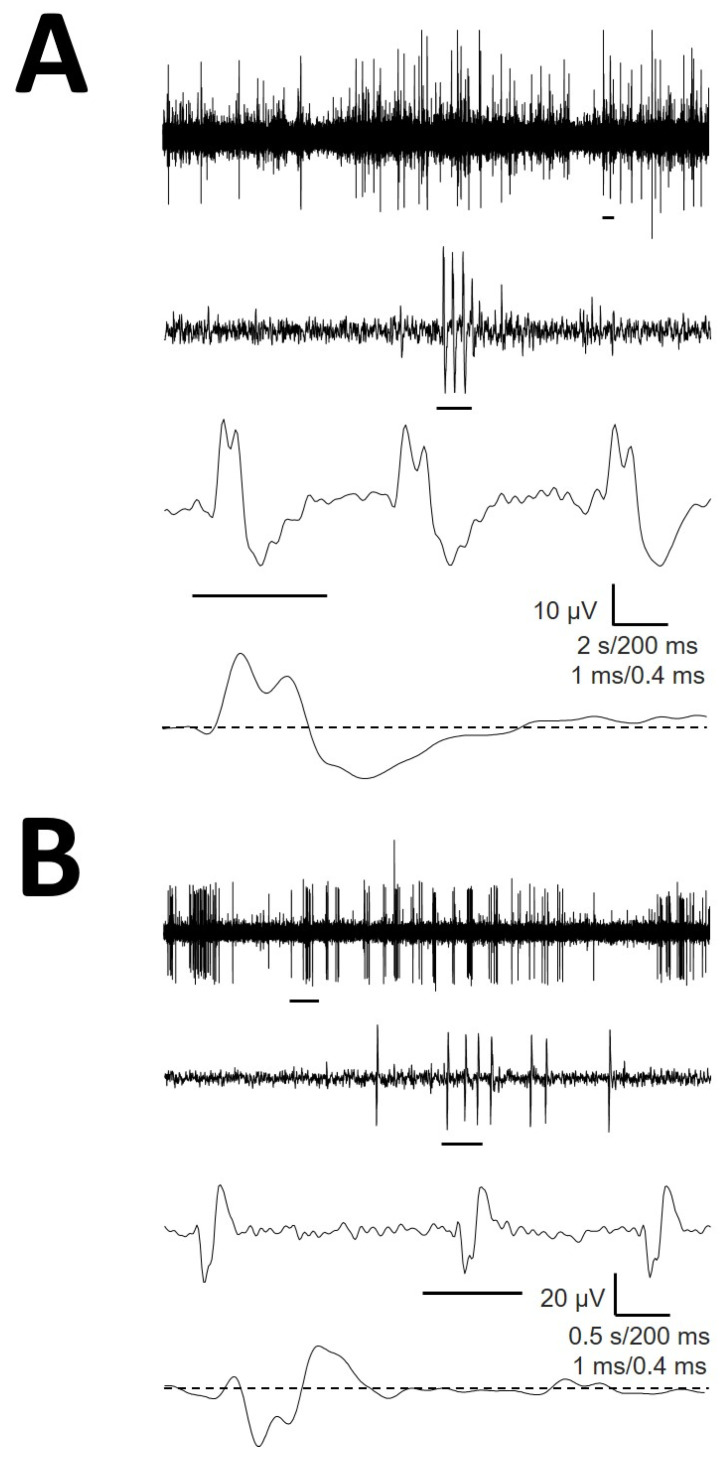
Atypical EAP obtained from raw recordings from (**A**) patients #1 (**B**) and #3. A small horizontal line under each trace is expanded in the trace immediately below. Values for horizontal calibration bars correspond to upper to lower traces.

**Table 1 brainsci-12-00043-t001:** Absolute frequency and percentage (between brackets) computed for the different types of mAP at the three regions.

Region		Type of mAP					
	P1N1	P1P2N	N1P1N2	P1N1	P1N1P2	amAP	Total
uPMH	25 (36)	9 (13)	23 (33)	2 (3)	4 (6)	7 (10)	70 (100)
PMH	12 (21)	12 (21)	12 (21)	5 (9)	4 (7)	11 (20)	56 (100)
lPMH	18 (35)	8 (15)	18 (35)	4 (8)	1 (2)	3 (6)	52 (100)

**Table 2 brainsci-12-00043-t002:** Properties of types of mAP across the three regions.

mAP	Region	V_FP_ (µV)	V_Dep_ (µV)	V_Rep_ (µV)	V_mAP_ (µV)	d_FP_ (ms)	d_Dep_ (ms)	d_Rep_ (ms)	d_mAP_ (ms)	dV_max_ (mV/s)	dV_min_ (mV/s)
	uPMH	-	45.5 ± 3.1	−25.5 ± 1.6	71.4 ± 3.7	-	0.43 ± 0.02 **	1.16 ± 0.08	1.59 ± 0.08	3.4 ± 0.3	−3.3 ± 0.3
P1N1	PMH	-	50.6 ± 9.0	−33.7 ± 5.1	84.3 ± 13.8	-	0.47 ± 0.03 *	1.20 ± 0.08 *	1.66 ± 0.08	3.5 ± 0.5	−3.8 ± 0.8
	lPMH	-	43.5 ± 2.2	−28.4 ± 2.1	71.9 ± 3.8	-	0.47 ± 0.02 *	0.94 ± 0.03 *	1.41 ± 0.02	2.7 ± 0.2	−3.4 ± 0.3
	uPMH	11.3 ± 3.0	41.8 ± 4.1	−27.4 ± 2.9	69.3 ± 6.8	0.25 ± 0.02	0.38 ± 0.02	0.91 ± 0.11 **	1.54 ± 0.12 **	2.9 ± 0.4	−3.7 ± 0.5
P1P2N1	PMH	18.8 ± 3.4	50.3 ± 7.1	−31.7 ± 3.8	82.0 ± 10.5	0.28 ± 0.02	0.37 ± 0.01	1.39 ± 0.09 *	2.04 ± 0.10 *	2.4 ± 0.5	−3.5 ± 0.5
	lPMH	8.8 ± 1.4	38.9 ± 7.1	−32.4 ± 5.8	71.2 ± 12.8	0.25 ± 0.03	0.38 ± 0.02	1.04 ± 0.09 *	1.64 ± 0.11 *	2.2 ± 0.4	−3.0 ± 0.7
	uPMH	−9.0 ± 1.1	54.2 ± 7.0	−33.2 ± 2.6	87.4 ± 9.1	0.28 ± 0.03	0.39 ± 0.02	1.11 ± 0.10	1.79 ± 0.10	4.6 ± 0.6	−4.1 ± 0.4
N1P1N2	PMH	−9.6 ± 1.8	45.8 ± 7.4	−25.2 ± 2.1	71.0 ± 8.9	0.23 ± 0.02	0.36 ± 0.02	1.33 ± 0.14	1.92 ± 0.14 *	4.4 ± 0.6	−3.3 ± 0.5
	lPMH	−7.7 ± 0.9	50.5 ± 5.5	−31.7 ± 3.2	82.2 ± 8.2	0.27 ± 0.06	0.33 ± 0.02	0.98 ± 0.05	1.57 ± 0.11 *	4.9 ± 0.6	−5.0 ± 0.8
	uPMH	−9.0	−43.1	22.4	65.6	0.05	0.57	1.19	1.81	3.0	−3.0
N1P1	PMH	−0.7 ± 1.0	−47.9 ± 6.9	35.1 ± 5.7	83.0 ± 12.5	0.04 ± 0.04	0.69 ± 0.06	1.45 ± 0.17	2.14 ± 0.24	2.5 ± 0.3	−2.2 ± 0.4
	lPMH	−1.4 ± 0.8	−36.5 ± 3.9	27.4 ± 2.2	63.9 ± 5.8	0.05 ± 0.03	0.62 ± 0.09	1.34 ± 0.05	1.99 ± 0.10	2.4 ± 0.5	−1.8 ± 0.1
	uPMH	5.6 ± 1.5	−29.2 ± 5.8	25.1 ± 2.5	54.4 ± 6.8	0.42 ± 0.09	0.51 ± 0.09	1.35 ± 0.14	2.29 ± 0.17	1.8 ± 0.7	−2.4 ± 0.7
P1N1P2	PMH	0.5	−58.5	38.1	96.6	0.09	0.70	1.53	2.23	2.61	−2.2
	lPMH	5.5	−33.1	33.8	66.9	1.15	0.53	1.20	1.73	3.0	−2.1

Asterisks of the same color (red or blue) indicate a significant difference for paired post-hoc Dunn’s test.

**Table 3 brainsci-12-00043-t003:** Comparison between thalamic cells (obtained from a previous work) and hypothalamic cells.

mAP	Region	N	dV_max_ (mV/s)	dV_min_ (mV/s)	V_FP_ (µV)	V_Dep_ (µV)	V_Rep_ (µV)	V_mAP_ (µV)	d_FP_ (ms)	d_Dep_ (ms)	d_Rep_ (ms)	d_mAP_ (ms)	Freq_cell_ (Hz)
	Thalam.	70	4.6 ± 0.4	−4.0 ± 0.4	-	55.8 ± 4.6	−21.9 ± 2.9	78.8 ± 7.2	-	0.43 ± 0.02	1.53 ± 0.9	1.99 ± 0.8	0.6 ± 0.1
P1N1	Hypoth.	55	3.2 ± 0.2	−3.5 ± 0.2	-	45.9 ± 2.5	−28.4 ± 1.5	74.3 ± 3.6	-	0.45 ± 0.01	1.10 ± 0.04	1.55 ± 0.04	1.3 ± 0.3
	Signif.		n.s.	n.s.		n.s.	0.001	n.s.		0.05	0.001	0.001	0.001
	Thalam.	811	6.6 ± 0.2	−6.3 ± 0.2	18.2 ± 0.6	95.3 ± 2.2	−53.4 ± 1.1	148.8 ± 3.3	0.11 ± 0.00	0.38 ± 0.00	1.78 ± 0.01	2.83 ± 1.1	1.1 ± 0.1
P1P2N1	Hypoth.	29	2.5 ± 0.2	−3.4 ± 0.3	13.7 ± 1.9	44.5 ± 3.8	−30.6 ± 2.3	75.1 ± 5.9	0.27 ± 0.01	0.37 ± 0.01	1.15 ± 0.07	1.77 ± 0.07	1.6 ± 0.7
	Signif.		0.001	0.001	n.s.	0.001	0.001	0.001	0.001	n.s.	0.001	0.001	n.s.
	Thalam.	216	8.2 ± 0.4	−6.1 ± 0.3	−15.6 ± 0.7	85.2 ± 4.0	−36.9 ± 1.9	122.1 ± 5.8	0.16 ± 0.00	0.36 ± 0.01	1.14 ± 0.01	1.66 ± 0.03	1.7 ± 0.3
N1P1N2	Hypoth.	53	4.7 ± 0.3	−4.2 ± 0.3	−8.7 ± 0.7	51.0 ± 3.9	−30.9 ± 1.7	81.9 ± 5.2	0.27 ± 0.02	0.34 ± 0.01	1.12 ± 0.06	1.74 ± 0.07	1.3 ± 0.3
	Signif.		0.001	0.001	0.001	0.001	n.s.	0.01	0.001	0.001	0.001	n.s.	0.001
	Thalam.	17	8.5 ± 0.9	−8.6 ± 0.8	6.9 ± 1.0	−102.6 ± 10.9	46.4 ± 6.1	149.0 ± 16.2	0.08 ± 0.03	0.46 ± 0.06	1.98 ± 0.07	2.65 ± 0.12	2.5 ± 0.1
P1N1P2	Hypoth.	9	2.3 ± 0.3	−2.3 ± 0.3	4.5 ± 1.1	−36.5 ± 3.7	26.1 ± 1.3	62.6 ± 4.4	0.28 ± 0.07	0.56 ± 0.05	1.26 ± 0.08	2.06 ± 0.15	1.0 ± 0.6
	Signif.		0.001	0.001	n.s.	0.001	0.01	0.001	0.001	0.001	0.001	n.s.	0.01

Hypoth. = hypothalamic; Thalam. = thalamic; Signif. = statistical significance, lower than the value indicated for the Mann-Whitney U test.

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
