# Peer review of "Neurophysiological Characterization of Posteromedial Hypothalamus in Anaesthetized Patients"

_brainsci, 2021, doi:10.3390/brainsci12010043_

Round 1

Reviewer 1 Report

  • The authors may wish to place their MATLAB code online to help boost reproducibility.
  • Forgive me if I’m not familiar with the terms typically used in the field, but would it not make more sense to calculate BI as the proportion of all spikes that were associated with ISI < 20ms, rather than dividing the # of ISI < 20ms by the # of ISI > 20 ms? Similar claims could be make about PI and PR.
  • Kurtosis was calculated. Why not skewness and variance?
  • Why was p set to 0.05. Was there a justification for not correcting for multiple comparisons?
  • Summing the number of positive (158) and atypical (21) mAPs = 179 traces, however there were only 178 traces. Was one (or more) of the positive traces atypical? If so, does this have an impact on the authors conclusions?
  • There are numerous sub-optimal grammatical choices – e.g., some paragraphs are only a single sentence, whereas others take almost a full page.

Author Response

  • The authors may wish to place their MATLAB code online to help boost reproducibility.

We have added an availability sentence at the end of the manuscript.

  • Forgive me if I’m not familiar with the terms typically used in the field, but would it not make more sense to calculate BI as the proportion of all spikes that were associated with ISI < 20ms, rather than dividing the # of ISI < 20ms by the # of ISI > 20 ms? Similar claims could be make about PI and PR.

This is a very interesting question. We had two reasons to select the definition of BI (and the rest): 1) to maintain consistency with previous literature used by our group and others (Favre et al, 1999; 2004) in previous works and 2) because in fact, what we addressed with this measure was not the percentage of cells with a defined ISI, but a ratio trying to capture the essence of a burst, defined by groups of short ISI separated by longer periods.

  • Kurtosis was calculated. Why not skewness and variance?

We have used kurtosis (K) as a measure of the homogeneity of the group. Therefore K > 7 indicates the presence of outliers that could be removed and K < 2 would suggest a mixed group. Therefore, 2<K<7 is the way to obtain homogenous groups. Skewness was addressing during measurement of normality by Kolmogorov-Smirnov test. Variance of the group is related to SEM, the dispersion measure used.

  • Why was p set to 0.05. Was there a justification for not correcting for multiple comparisons?

The significance at p = 0.05 is the most widely used in medical and biological sciences. Most of the comparisons at the manuscript were pairwise (tables 2 and 3). In those case where several elements of different groups were compared (data from table 1) Chi-squared was computed with and without the Yate’s correction with similar results. We have added a sentence at lines 223-224 and 274-275.

  • Summing the number of positive (158) and atypical (21) mAPs = 179 traces, however there were only 178 traces. Was one (or more) of the positive traces atypical? If so, does this have an impact on the authors conclusions?

Thanks for this observation. There was a mistake in the number of P1N1P2 cells, and we erroneously wrote 10 instead of 9, the true figure. We have reviewed all the manuscript and modified accordingly the small number of mistakes found. Of course, this typo does not affect any of the conclusions rosed at the manuscript.

  • There are numerous sub-optimal grammatical choices – e.g., some paragraphs are only a single sentence, whereas others take almost a full page.

We are conscious that our English’s style is not as good as English natives. Therefore, we sent the manuscript to be edited by professional native editors (AJE). We have attached the certificate with the original version.

Reviewer 2 Report

The paper presents an interesting article about the characterization of the neurophysiological properties of the Posteromedial Hypothalamus in 4 anesthetized patients.

The article is generally clear and well written. While I have no comments on the recoding procedure, I have some minor suggestions for the presentation and reporting of the findings.

Comments

Given that the line number is resetted after page 9, I have divided the notes before/after page 9.

Before page 9:

  • Given that the focus of the article is not Erethism, I would suggest the authors to shift the first paragraph of the introduction after the second one, or as an example of possible application. Currently it's a bit complicated to follow the first paragraph after the abstract and title.
  • Line 54. "in the hypothalamus, these stimuli are not useful". Why is that so? Authors should consider extending the sentence to clarify the meaning of this point.
  • Line 64. Flor clarity, I would suggest the authors to move the reference to [11] to the previous sentence, where the study is mentioned. 
  • Line 110. I would suggest the authors to describe briefly the procedure of [20] and then indicate the details are in [20] to make the paper easier to read without switching from a reference to the other.
  • Figure 1. I would suggest the authors to increase the size of the figures as some elements are difficult to interpret (e.g. axis labels of 1C, plot of 1F).
  • Line 171. I would suggest the authors to use the word following instead of next to avoid possible misinterpretation.  
  • Line 192. I think this should be Figure 1F.
  • Line 205. I think the comma at the end of the line should be a dot.

After page 9:

  • Line 32. How is atypical defined? visual inspection? If so how did the authors proceeded in identifying atypical structures? Was there more than one coder or has a statistical test been used?
  • Line 85.  I think "our case" should be "In our case".

Author Response

The paper presents an interesting article about the characterization of the neurophysiological properties of the Posteromedial Hypothalamus in 4 anesthetized patients.

The article is generally clear and well written. While I have no comments on the recoding procedure, I have some minor suggestions for the presentation and reporting of the findings.

We acknowledge the reviewer’s opinion and we feel encouraged.

Comments

Given that the line number is resetted after page 9, I have divided the notes before/after page 9.

Sorry for this error. We have corrected in this new version.

Before page 9:

  • Given that the focus of the article is not Erethism, I would suggest the authors to shift the first paragraph of the introduction after the second one, or as an example of possible application. Currently it's a bit complicated to follow the first paragraph after the abstract and title.

We have modified the Introduction according to this suggestion.

  • Line 54. "in the hypothalamus, these stimuli are not useful". Why is that so? Authors should consider extending the sentence to clarify the meaning of this point.
  • We have added a sentence clarifying this aspect.
  • Line 64.Flor clarity, I would suggest the authors to move the reference to [11] to the previous sentence, where the study is mentioned. 
  • We have done.
  • Line 110.I would suggest the authors to describe briefly the procedure of [20] and then indicate the details are in [20] to make the paper easier to read without switching from a reference to the other.

We have added this paragraph “Briefly, to do that, we considered the real x_r and y_r coordinates measured by MRI performed one month after surgery and represented (scaled according to AC-PC distance) them on the last axial plane of the SW map, including the floor of the thalamus (z = 0). We defined this point as the real end-point r ⃗_ep=(x_r,y_r,0). The angles used during MER trajectory (ring and arch) were the same as those measured from the final DBS electrode during the post-op MRI. Starting from r ⃗_ep and considering the angles, we were able to reconstruct the real trajectory of the electrode in a three-dimensional space in 1 mm intervals. To select the nearest SW plane, we used only the trajectory corresponding to the centre electrode, which was the same electrode considered during the planning process. However, we reconstructed trajectories for all the electrodes. Frontal (F), sagittal (S) and axial (A) planes of SW atlas were digitized in individual jpg files and every file was identified by the distance to inter-commisural mid-point. For every dimension, SW is a set of irregular and discrete series of planes (20 for the frontal, from Fa165 to Fp155, 20 for the axial, from Av085 to Ad160 and 17 for the sagittal, from S015 to S275). For every distance to r ⃗_ep, it’s easy to identify the coordinates where the central electrode is placed (x_ce,y_ce,z_ce). Then, we obtain the orthogonal planes including this point and, for every plane, we select from the series of jpg files, the closest one. This algorithm was implemented in a MATLAB® script. For every plane, we plotted the projection of all the microelectrodes”

  • Figure 1.I would suggest the authors to increase the size of the figures as some elements are difficult to interpret (e.g. axis labels of 1C, plot of 1F).

Modified

  • Line 171. I would suggest the authors to use the word following instead of next to avoid possible misinterpretation.  

Done

  • Line 192.I think this should be Figure 1F.

Sorry, we don’t identify what do you mean

  • Line 205. Ithink the comma at the end of the line should be a dot.

Done

After page 9:

  • Line 32. How is atypical defined? visual inspection? If so how did the authors proceeded in identifying atypical structures? Was there more than one coder or has a statistical test been used?

The reviewer is right, and the definition was visual and not numerical. In fact, we state “These cells had amplitudes, durations, or numbers of phases that clearly differed from the remainder of the mAP”. These anomalous features are clearly visible from figs 2 and 4. Obviously we would need a more rigorous characterization but, as also stated in Discussion “The detailed description of these cells is beyond the scope of this work”.

  • Line 85.  I think "our case" should be "In our case".

Done

Reviewer 3 Report

In this manuscript, the authors provide a neurophysiological characterization of the posteromedial hypothalamus (PMH) and neighboring regions. Previously, authors had described basic electrophysiological properties of these neurons, such as power spectrum analysis of LFP, or the effect of stimulation on the firing pattern. Here, the authors provide an impressively detailed analysis of extracellular action potential. They suggest that neurons of the PMH have a distinctive neurophysiological signature, including a higher prevalence of neurons with P1P2N1 and an “atypical” action potential pattern. In addition, for a given action potential pattern, neurons of the PMH are different from thalamic neurons for derivatives, amplitude or duration of repolarization, and depolarization phases. The manuscript is well-written, the methods are well-described, and the results are novel and interesting.

I have some specific comments:

  • Line 31: I am doubtful about the terminology “erethism”. A PubMed search gives 48 results, all about mercury poisoning. And the reference given for the word erethism [ref-1] does not mention this word. It is not mentioned further in the manuscript. Why not simply “intractable aggressivity” or something similar?
  • Line 90: Recordings are made 10mm above and 3mm below the target. In the results section, the authors only mention distinctions between PMH and regions “above” or “below”. Authors should be more specific about these regions – 3mm below might be in the red nucleus, and 10mm above is likely in the ventral thalamus. They might want to add a figure highlighting these regions.
  • The offline signal processing to obtain the mean action potentials seems pretty tedious for it to be useful for intraoperative targeting. Have authors envisioned some automation of the signal processing? Otherwise, these analyses will have an impact in basic neurophysiology science, but not in clinical outcomes.
  • If I understand correctly, the classification of mean action potentials into subtypes (P1P2N1, N1P1N2, etc.) is made based on a subjective visual assessment of the processed and averaged signal. Given that some waveforms are somewhat similar, did authors compare assessments from different raters (inter-rater reliability)? It would be especially important for the “atypical” action potentials.
  • It is not clear to me what these “atypical” action potentials actually are. It is interesting that more of these are identified in the PMH compared to the thalamus. My impression is that such detailed analysis of subtle voltage variations is limited by the nature of the recording (extracellular); extracellular recordings are limited by variability of electrode location relative to nearby neurons. The way to clarify these results would be to perform intracellular recordings of PMH in animals, with a correlation of the histologic subtype with the neurophysiological properties.
  • There are over 50 acronyms in this article, which obscure the reading rather than simplify it. I suggest either reducing the number of acronyms or providing a table to define all the acronyms.

Overall, the manuscript is very interesting and well-written, and I believe it should be accepted pending minor revisions.

Author Response

In this manuscript, the authors provide a neurophysiological characterization of the posteromedial hypothalamus (PMH) and neighboring regions. Previously, authors had described basic electrophysiological properties of these neurons, such as power spectrum analysis of LFP, or the effect of stimulation on the firing pattern. Here, the authors provide an impressively detailed analysis of extracellular action potential. They suggest that neurons of the PMH have a distinctive neurophysiological signature, including a higher prevalence of neurons with P1P2N1 and an “atypical” action potential pattern. In addition, for a given action potential pattern, neurons of the PMH are different from thalamic neurons for derivatives, amplitude or duration of repolarization, and depolarization phases. The manuscript is well-written, the methods are well-described, and the results are novel and interesting.

We are very grateful to the reviewer by this opinion and by its rigorous revision.

I have some specific comments:

  • Line 31: I am doubtful about the terminology “erethism”. A PubMed search gives 48 results, all about mercury poisoning. And the reference given for the word erethism [ref-1] does not mention this word. It is not mentioned further in the manuscript. Why not simply “intractable aggressivity” or something similar?

In fact, is a little old nomenclature. We follow the reviewer’s suggestion and remove erethism.

  • Line 90: Recordings are made 10mm above and 3mm below the target. In the results section, the authors only mention distinctions between PMH and regions “above” or “below”. Authors should be more specific about these regions – 3mm below might be in the red nucleus, and 10mm above is likely in the ventral thalamus. They might want to add a figure highlighting these regions.

The reviewer is right, and this description can be misleading. The upper limit (10 mm) was a mistake. We have added this sentence to clarify: “Neuronal recordings (…) were obtained beginning 5 mm above the target and progressing in steps of 0.5 mm. MERs (…) were obtained until 2-3 mm below the theoretical target. However, the analysis for this work were restricted only to the regions near the place of maximum physiological response to electrical stimulation (see below).”

  • The offline signal processing to obtain the mean action potentials seems pretty tedious for it to be useful for intraoperative targeting. Have authors envisioned some automation of the signal processing? Otherwise, these analyses will have an impact in basic neurophysiology science, but not in clinical outcomes.

The reviewer is right again. However, the automation would be easy during on-line recording if we use mAP described as templates to classify extracellular action potentials. This way would be fast enough to be used during surgery. The practical problem is to obtain digital recordings from commercial equipment’s, because the most time-consuming process is the ASCII file conversion.

  • If I understand correctly, the classification of mean action potentials into subtypes (P1P2N1, N1P1N2, etc.) is made based on a subjective visual assessment of the processed and averaged signal. In fact, sorting EAP into mAPs was an automatic process, as can be observed from Analysis of Extracellular Action Potentials. At point 2 it’s described the definition of positive or negative first-phase according to the thresholds defined. We have added an accessory figure showing this at Appendix.
  • Given that some waveforms are somewhat similar, did authors compare assessments from different raters (inter-rater reliability)? It would be especially important for the “atypical” action potentials. In table 2 we addressed the similitude between different mAPs at the three regions. We showed that the mAPs were similar at all the three regions. Atypical cells were visually clearly different from canonical forms. A numerical characterization would imply an overwhelming work exceeding the length of a typical paper. We plan to characterize these cells in a new work.
  • It is not clear to me what these “atypical” action potentials actually are. It is interesting that more of these are identified in the PMH compared to the thalamus. My impression is that such detailed analysis of subtle voltage variations is limited by the nature of the recording (extracellular); extracellular recordings are limited by variability of electrode location relative to nearby neurons. The way to clarify these results would be to perform intracellular recordings of PMH in animals, with a correlation of the histologic subtype with the neurophysiological properties.

We absolutely agree. However, it is interesting to keep in mind that the same variability of electrode location relative to nearby neurons would be observed at thalamus or any other place of the brain recorded during DBS but nevertheless, these cells have not been observed.

  • There are over 50 acronyms in this article, which obscure the reading rather than simplify it. I suggest either reducing the number of acronyms or providing a table to define all the acronyms.

Thank you again by this suggest. We have added a table at Appendix.

Overall, the manuscript is very interesting and well-written, and I believe it should be accepted pending minor revisions.

Round 2

Reviewer 1 Report

The authors have addressed my concerns.